# Four Unique Genetic Variants in Three Genes Account for 62.7% of Early-Onset Severe Retinal Dystrophy in Chile: Diagnostic and Therapeutic Consequences

**DOI:** 10.3390/ijms25116151

**Published:** 2024-06-03

**Authors:** Rene Moya, Clémentine Angée, Sylvain Hanein, Fabienne Jabot-Hanin, Josseline Kaplan, Isabelle Perrault, Jean-Michel Rozet, Lucas Fares Taie

**Affiliations:** 1Department of Ophthalmology, Hospital del Salvador, Universidad de Chile, Santiago 7500922, Chile; rmoyab@yahoo.com; 2Laboratory of Genetics in Ophthalmology (LGO), INSERM UMR1163, Institute of Genetic Diseases, Imagine and Paris Descartes University, 75015 Paris, Franceisabelle.perrault@inserm.fr (I.P.); 3Bioinformatic Platform, INSERM UMR1163, Institute of Genetic Diseases, Imagine and Paris Descartes University, 75015 Paris, France

**Keywords:** Leber congenital amaurosis (LCA), early-onset retinal dystrophy (EOSRD), Chile

## Abstract

Leber congenital amaurosis (LCA)/early-onset severe retinal dystrophy (EOSRD) stand as primary causes of incurable childhood blindness. This study investigates the clinical and molecular architecture of syndromic and non-syndromic LCA/EOSRD within a Chilean cohort (67 patients/60 families). Leveraging panel sequencing, 95.5% detection was achieved, revealing 17 genes and 126 variants (32 unique). *CRB1*, *LCA5*, and *RDH12* dominated (71.9%), with *CRB1* being the most prevalent (43.8%). Notably, four unique variants (*LCA5* p.Glu415*, *CRB1* p.Ser1049Aspfs*40 and p.Cys948Tyr, *RDH12* p.Leu99Ile) constituted 62.7% of all disease alleles, indicating their importance for targeted analysis in Chilean patients. This study underscores a high degree of inbreeding in Chilean families affected by pediatric retinal blindness, resulting in a limited mutation repertoire. Furthermore, it complements and reinforces earlier reports, indicating the involvement of *ADAM9* and *RP1* as uncommon causes of LCA/EOSRD. These data hold significant value for patient and family counseling, pharmaceutical industry endeavors in personalized medicine, and future enrolment in gene therapy-based treatments, particularly with ongoing trials (*LCA5*) or advancing preclinical developments (*CRB1* and *RDH12*).

## 1. Introduction

Inherited Retinal Dystrophies (IRDs) include a large and heterogeneous group of disorders in which rod and/or cone photoreceptors degenerate in a diffuse or regionalized manner, causing variable degrees of visual handicap [1]. Leber congenital amaurosis (LCA, MIM #204000) and early-onset severe retinal dystrophy (EOSRD) are the earliest and most severe of these diseases, manifesting in blindness or profound vision deficiency at birth or within the very first years of life [2]. The initial aspect of the retina is typically unremarkable, but electroretinography (ERG) shows undetectable or deeply altered scotopic and photopic responses, in keeping with a profound dysfunction of rod and cone photoreceptors [3]. LCA/EOSRD are the most common causes of blindness or profound vision impairment in childhood, affecting 20% of children in schools for the blind [4]. They are typically autosomal recessive diseases, although some dominant forms exist. They can occur as non-syndromic disorders or as the initial symptom in a range of devastating syndromes [2], in particular ciliopathies [5,6,7,8,9]. Some twenty and thirty genes have been involved in non-syndromic and syndromic LCA/EOSRD diseases, and it is known that there is little genetic overlap between the two presentations [5,9]. Identifying the underlying genetic defect can aid in early differential diagnosis and tailor-made extraocular function follow-up, adding further value to the molecular diagnosis of these diseases, whose importance has increased with therapies on the market or in advanced clinical trials [10].

Here, we report the results of panel-based molecular diagnosis in a Chilean cohort, which provide a comprehensive representation of the genetic landscape of LCA/EORD in Chile.

## 2. Results

### 2.1. Data Summary

Pathogenic or likely pathogenic variants were detected in all but three sporadic cases, achieving a diagnostic rate exceeding 95% (64/67 individuals within 57/60 families from all over the country; Table 1). The genotypes of the subjects aligned with their diagnoses in over 90% of cases. Among them, 55 individuals (49 families) with non-syndromic LCA or EOSRD carried recessive variants in *CRB1*, *CEP290*, *GUCY2D*, *LCA5*, *NMNAT1*, *RDH12*, *RPGRIP1*, *SPATA7*, or *TULP1* or a dominant variant in *CRX*. Additionally, three individuals (two families) with syndromic presentations had mutations in *ALMS1* or *IQCB1*. Less than 10% of the individuals (six sporadic cases) exhibited mutations associated with other recessive or dominant IRD genes *(ADAM9*, *NR2E3*, *RAB28*, and *RP1* and *PRPF31*, respectively). Approximately two-thirds of the cases (42 individuals within 38 families) displayed homozygous genotypes, indicating a high level of population inbreeding (Figure 1A).

In total, we identified 124 recessive and 2 dominant disease alleles. Recessive alleles included missense (55/124), frameshifting indel (34/124), nonsense (23/124), large deletion (4/124), canonical or non-canonical splice-site (4/124), large duplication (2/124), and in-frame deletion (2/124) changes (Table 1). Recessive alleles included 30 unique variants, 14 of which were novel. *CRB1* accounted for the majority of disease alleles (56/124), followed by *RDH12* (18/124) and *LCA5* (18/124), while *ADAM9*, *ALMS1*, *CEP290*, *GUCY2D*, *IQCB1*, *NMNAT1*, *NR2E3*, *RAB28*, *RP1*, *RPGRIP1*, *SPATA7*, and *TULP1* each contributed one or two cases (Figure 1B). The two dominant alleles comprised previously reported frameshift and nonsense variants in *CRX* [11] and *PRPF31* [12], respectively (Table 1). Samples from family members were available in 30 cases (46.8%) for cosegregation, which allowed confirming biparental transmission in recessive cases.

### 2.2. Individuals with Mutations in Established Genes Associated with LCA or EOSRD

*CRB1*. Twenty-eight individuals from twenty-four families, constituting approximately 50% of those with LCA/EOSRD gene variants (28/55 within 24/49 families), displayed *CRB1* variants (Table 1). Among these, eight were unique, consisting of three previously reported and five novel variants. Novel variants included nonsense, frameshift, and consensus splice-site changes. Two of these were found in two apparently unrelated individuals each, while the remaining three were unique to individual cases. Twenty-four of the twenty-eight individuals carried the previously reported c.2843G>A (p.Cys948Tyr) and/or c.3110_3143dup (p.Ser1049Aspfs*40) mutations [13], mostly identified in homozygosity or in compound heterozygosity with each other. Other cases were associated with either another reported mutation or a novel change. Together, these two mutations accounted for 82.7% of *CRB1* disease alleles (48/56:31/56 and 17/56, respectively), establishing them as the primary contributors to LCA/EOSRD in the pediatric IRD population in Chile. Haplotype analysis at the *CRB1* locus in eight individuals carrying the c.3110_3143dup mutation allowed the identification of a common haplotype of 110.5 kb, which could have appeared in a common ancestor 11 generations ago (confidence interval: 7–9 generations). The four remaining subjects out of the twenty-eight were compound heterozygous for a reported mutation and a novel variant, except for one subject who carried two novel changes. All individuals displayed the typical *CRB1*-LCA phenotype, generally diagnosed before reaching the age of 1 year (Table 2). Nystagmus typically emerged in the first year, with over half exhibiting Franceschetti’s oculo-digital signs. Nearly universal night blindness, often with photoaversion, was observed from early childhood. Moderate to high hyperopia was common, while myopia was infrequent. Fundus exams revealed macular and peripheral atrophy, with most exhibiting pseudocoloboma, nummular RPE pigmentation deposits, and Preserved Para-arteriolar Retinal Pigment Epithelium (PPRPE). Vision varied from low to absent, correlating with age (4–54 years).

*LCA5*. Nine individuals (16.4%, 9/55) from eight families presented *LCA5* mutations (Table 1). All of them carried the reported c.1243G>T (p.Glu415*) nonsense mutation, either in homozygosity or compound heterozygosity with the novel and recurring c.1569_1582del (p.His523Glnfs16) change (7/9 and 2/9 individuals, respectively). In total, the c.1243G>T (p.Glu415*) mutation constituted 89% (16/18) of *LCA5* disease alleles. Haplotype analysis at the *LCA5* locus in homozygous individuals identified a common haplotype of 442 kb, which could have arisen in a common ancestor five generations ago (confidence interval: 2–21 generations). The individuals typically displayed profound visual deficiency at or near birth, often accompanied by nystagmus and, in some cases, digito-ocular signs of Franceschetti. Early-onset night blindness was common, with photophobia in some cases. Vision ranged from infantile blindness to low but measurable BCVA at an advanced age. Individuals exhibited either myopia or hypermetropia, and fundus exams showed a spectrum of aspects, ranging from only mild peripheral changes to the advanced stage of Retinitis Pigmentosa (RP). This significant variation was observed regardless of the individual’s age (ranging from 6 to 58) and genotype (Table 2).

*RDH12*. Another nine individuals from eight families carried *RDH12* disease alleles, which consisted of three unique and previously reported variants (Table 1). The majority of mutant alleles (15/18) were represented by the c.295C>A (p.Leu99Ile) substitution [14], identified in eight out of the nine individuals. Among these, seven were homozygous, while one carried the change in compound heterozygosity with the c.716G>T (p.Arg239Leu) [14] substitution. The remaining subject among the nine was homozygous for a 1-bp duplication affecting the same Arg239 residue (c.715dup, p.Arg239Profs*34) [15]. Individuals with the p.Leu99Ile substitution in homozygosity displayed variable onset of visual symptoms (0–7 years) without reported nystagmus or digito-ocular signs of Franceschetti. In their most recent examination (6–36 years), they frequently reported nyctalopia, photophobia, or both. The majority had mild hyperopia, while one had mild myopia. Vision ranged from blindness to low but measurable Best-Corrected Visual Acuity (BCVA), with the youngest having the best BCVA and the eldest exhibiting the lowest visual function. All individuals but the youngest exhibited typical *RDH12*-associated widespread pigmentary retinopathy with early-onset central involvement [15]. Individuals with variants affecting Arg239 presented with a similar disease manifestation but at a younger age and exhibited nystagmus (Table 2).

*CEP290*. Two individuals (3.6%, 2/55) carried variants in *CEP290*, known for its role in non-syndromic IRD or syndromic ciliopathies with IRD (Table 1). In one case, a nine-year-old girl carried the non-syndromic LCA-causing hypomorphic founder c.2991+1655A>G mutation [16] in homozygosity. Unexpectedly, she showed symptoms of EOSRD instead of the anticipated phenotype of LCA. The onset of the disease occurred at 18 months, characterized by the absence of nystagmus, oculo-digital signs, or photophobia. At the age of 9 years, ERG responses were severely altered yet detectable, and she displayed moderate hyperopia, mild peripheral changes at the fundus, and low but measurable BCVA. The second individual carried a missense variant of uncertain significance together with a LCA-causing mutation (c.38T>A (p.Val13Asp) and c.7341dup (p.Leu2448Thrfs*8), respectively). This girl manifested a retinal disease consistent with *CEP290* involvement, supporting the pathogenicity of the missense change. She displayed nystagmus and the oculo-digital sign of Franceschetti at birth, with her visual function limited to light perception and minor fundus changes (Table 2) typical of *CEP290* at the age of 8 years [17]. Given that *CEP290* missense changes are predicted to be hypomorphic, it is unlikely that the subject will experience extraocular involvement.

*NMNAT1*. Two individuals (3.6%, 2/55) had reported *NMNAT1* changes, both sharing the recurrent c.769G>A (p.Glu257Lys) missense mutation [18] (Table 1). One had it in combination with a frameshift (c.364del (p.Arg122Glyfs20) [14,18]), while the other had it with nonsense (c.507G>A (p.Trp169*) [12,18]) mutations. The p.Glu257Lys substitution is recognized as hypomorphic, leading to LCA when combined with a severe mutation in trans [19]. In line with their genotype, the two individuals, aged 3 and 35 years, presented with severe visual impairment at birth, characterized by nystagmus and digito-ocular signs of Franceschetti. The youngest subject displayed the typical *NMNAT1*-associated central pseudocoloboma and peripheral atrophy at the fundus [18], while a bilateral cataract impeded fundus examination in the elder individual (Table 2).

*GUCY2D*, *RPGRIP1*, *SPATA7*, *TULP1*, *CRX*. Each of these genes was involved in unique cases, individually accounting for 1.8% (1/55) of the total cases. *GUCY2D*, *SPATA7*, and *CRX* variants were reported previously, while *RPGRIP1* and *TULP1* changes were novel (Table 1).

*GUCY2D* mutations were found in a 3-year-old individual who exhibited typical features associated with GUCY2D-related disease at this age, namely blindness from birth, nystagmus, digito-ocular signs of Franceschetti, photophobia, LP, high hyperopia, and an unremarkable fundus appearance (Table 2).

A dominant *CRX* mutation (c.434del (p.Pro145Leufs*42)) was detected in a 31-year-old woman. She had a more severe retinal disease compared to the initially reported 15-year-old Japanese girl carrying the same mutation [20]. The woman exhibited profound visual deficiency with nystagmus from birth, childhood night blindness, and teenage photophobia, along with widespread pigmentary retinopathy (Table 2). In contrast, the 15-year-old girl had myopia, night blindness, very reduced but measurable BVCA along with paracentral scotoma, and peripheral visual field defect, in line with a relatively preserved central retina and marked peripheral atrophy at the fundus [20].

In the case of *SPATA7*, we found homozygosity for a nonsense mutation (c.1171C>T (p.Arg391*)), which had been reported previously in homozygosity without clinical details [21]. No disease onset information was available for the Chilean subject; however, by the age of 42 years, he displayed nystagmus, severe vision impairment (LP and NLP), and widespread pigmentary retinopathy (Table 2), suggesting a highly severe retinal disease.

Concerning *RPGRIP1*, we discovered homozygosity for a novel duplication spanning exon 10 to 19, estimated to be 16 to 18 kb in size—this is the largest *RPGRIP1* duplication reported to date. This finding was observed in a 27-year-old individual exhibiting a typical *RPGRIP1*-associated phenotype [22], including severe visual impairment within the first year of life, nystagmus, significantly impaired vision, and pigmentary retinopathy. Notably, the individual displayed unilateral myopia and night blindness, in contrast to the moderate hyperopia and photoaversion typically associated with *RPGRIP1* mutations (Table 2).

We found homozygosity for a novel *TULP1* c.1149C>A (p.Asp383Glu) missense substitution classified as pathogenic in Varsome due to multiple lines of computational evidence, including the report of an RP-causing missense mutation affecting the same residue [23]. Further supporting pathogenicity, the 4-year-old individual with the mutation exhibited typical features of *TULP1*-associated disease [24], including severe visual impairment in infancy with nystagmus, photoaversion, night blindness, vision limited to HM and CF, high myopia, and widespread pigmentary retinopathy (Table 2).

**Table 1 ijms-25-06151-t001:** Summary of pathogenic variants detected in the 64 Chilean patients. Prioritizing strategy for filtering pathogenic variants was performed with PolyDiag, a user-friendly interface designed for panel-based molecular diagnosis that integrates genomic information from the public Genome Aggregation Database (gnomAD v2.1; https://gnomad.broadinstitute.org, accessed on 2 February 2023), ClinVar archive of relationships between variations and phenotypes (v20200706 https://www.ncbi.nlm.nih.gov/clinvar, accessed on 14 August 2023), Human Genome Database (HGMD v2020.2, www.hgmd.cf.ac.uk accessed on 1 December 2022), and the prediction algorithms for scoring the deleteriousness of variants in the human genome Combined Annotation Dependent Depletion (v.1.6, CADD; www.cadd.gs.washington.edu), Gencode v19 (https://www.gencodegenes.org accessed on 2 February 2023), Polyphen2 (http://genetics.bwh.harvard.edu/pph2/, accessed on 2 February 2023) [25], SIFT (http://sift.jcvi.org, accessed on 2 February 2023) [26], and Mutation Taster2 (www.mutationtaster.org, accessed on 2 February 2023) [27]. Variants absent in HGMD and ClinVar were ranked on the basis of their frequency in gnomAD (minor allele frequency MAF < 1% and 0.1 for recessive and dominant genes, respectively) and their predicted consequence on the protein. Nonsense variants, indels, and duplications introducing premature termination codons, intronic variants disrupting canonical splice sites, and nonsynonymous single-nucleotide variants (SNVs) and intronic changes with highest CADD scores were given priority. Homozygous variants were inferred from read-depth analysis of the NGS panel data. Abbreviations are as follows: P, pathogenic; LP, likely pathogenic; VUS, uncertain significance. N.A, Not Available.

					Allele 1					Allele 2				
Family	Patient	Gene	MIM#	NM_#	Variant	Parental Origin	Exons	ACMG Category	HGMD Accession (Citation Numbers)	Variant	Parental Origin	Exons	ACMG Category	HGMD
1	FG313	*ADAM9*	602713	003816.3	c.333+2_1303del	p	5–12		Novel	c.333+2_1303del	(m)	5–12		Novel
2	FG297	*ALMS1*	606844	001378454.1	c.1092del (p.Asp365IlefsTer11)	m	5	P	Novel	c.1092del (p.Asp365IlefsTer11)	(p)	5	P	Novel
	FG283	*ALMS1*	606844	001378454.1	c.1092del (p.Asp365IlefsTer11)	m	5	P	Novel	c.1092del (p.Asp365IlefsTer11)	(p)	5	P	Novel
3	FG277	*CEP290*	610142	025114.4	c.2991+1655A>G	N.A	Intron 26	P	CS064383 (31)	c.2991+1655A>G	N.A	Intron 26	P	CS064383 (31)
4	FG393	*CEP290*	610142	025114.4	c.38T>A (p.Val13Asp)	p	2	VUS	Novel	c.7341dup (p.Leu2448ThrfsTer8)	(m)	54	P	CI062252 (5)
5	FG50	*CRB1*	604210	201253.3	c.750T>A (p.Cys250Ter)	m	3	P	CM2041497 (1)	c.798_799del (p.Ala267GlnfsTer18)	p	3	P	Novel
6	FG66	*CRB1*	604210	201253.3	c.2843G>A (p.Cys948Tyr)	m	9	P	CM992152 (35)	c.3110_3143dup (p.Ser1049AspfsTer40)	p	9	P	CN205417 (1)
	FG224	*CRB1*	604210	201253.3	c.2843G>A (p.Cys948Tyr)	m	9	P	CM992152 (35)	c.3110_3143dup (p.Ser1049AspfsTer40)	p	9	P	CN205417 (1)
7	FG112	*CRB1*	604210	201253.3	c.2843G>A (p.Cys948Tyr)	m	9	P	CM992152 (35)	c.3110_3143dup (p.Ser1049AspfsTer40)	p	9	P	CN205417 (1)
	FG113	*CRB1*	604210	201253.3	c.2843G>A (p.Cys948Tyr)	m	9	P	CM992152 (35)	c.3110_3143dup (p.Ser1049AspfsTer40)	p	9	P	CN205417 (1)
8	FG128	*CRB1*	604210	201253.3	c.2466G>A (p.Trp822Ter)	NA	7	P	Novel	c.2843G>A (p.Cys948Tyr)	N.A	9	P	CM992152 (35)
9	FG239	*CRB1*	604210	201253.3	c.2843G>A (p.Cys948Tyr)	p	9	P	CM992152 (35)	c.2843G>A (p.Cys948Tyr)	(m)	9	P	CM992152 (35)
10	FG362	*CRB1*	604210	201253.3	c.3110_3143dup (p.Ser1049AspfsTer40)	m	9	P	CN205417 (1)	c.3110_3143dup (p.Ser1049AspfsTer40)	p	9	P	CN205417 (1)
11	FG272	*CRB1*	604210	201253.3	c.3110_3143dup (p.Ser1049AspfsTer40)	NA	9	P	CN205417 (1)	c.3110_3143dup (p.Ser1049AspfsTer40)	N.A	9	P	CN205417 (1)
	FG365	*CRB1*	604210	201253.3	c.3110_3143dup (p.Ser1049AspfsTer40)	NA	9	P	CN205417 (1)	c.3110_3143dup (p.Ser1049AspfsTer40)	N.A	9	P	CN205417 (1)
	FG366	*CRB1*	604210	201253.3	c.3110_3143dup (p.Ser1049AspfsTer40)	NA	9	P	CN205417 (1)	c.3110_3143dup (p.Ser1049AspfsTer40)	N.A	9	P	CN205417 (1)
12	FG390	*CRB1*	604210	201253.3	c.3110_3143dup (p.Ser1049AspfsTer40)	m	9	P	CN205417 (1)	c.3110_3143dup (p.Ser1049AspfsTer40)	(p)	9	P	CN205417 (1)
13	FG432	*CRB1*	604210	201253.3	c.2843G>A (p.Cys948Tyr)	m	9	P	CM992152 (35)	c.2843G>A (p.Cys948Tyr)	p	9	P	CM992152 (35)
14	FG436	*CRB1*	604210	201253.3	c.2843G>A (p.Cys948Tyr)	NA	9	P	CM992152 (35)	c.2843G>A (p.Cys948Tyr)	N.A	9	P	CM992152 (35)
15	FG444	*CRB1*	604210	201253.3	c.2843G>A (p.Cys948Tyr)	m	9	P	CM992152 (35)	c.2843G>A (p.Cys948Tyr)	p	9	P	CM992152 (35)
16	FG456	*CRB1*	604210	201253.3	c.2843G>A (p.Cys948Tyr)	NA	9	P	CM992152 (35)	c.2843G>A (p.Cys948Tyr)	N.A	9	P	CM992152 (35)
17	FG231	*CRB1*	604210	201253.3	c.2843G>A (p.Cys948Tyr)	NA	9	P	CM992152 (35)	c.2843G>A (p.Cys948Tyr)	N.A	9	P	CM992152 (35)
18	FG395	*CRB1*	604210	201253.3	c.3110_3143dup (p.Ser1049AspfsTer40)	m	9	P	CN205417 (1)	c.750T>A (p.Cys250Ter)	(p)	3	LP	CM2041497 (1)
19	FG399	*CRB1*	604210	201253.3	c.2843G>A (p.Cys948Tyr)	m	9	P	CM992152 (35)	c.2291G>A (p.Arg764His)	p	7	P	CM130791 (9)
20	FG649	*CRB1*	604210	201253.3	c.653-1G>A	NA	Intron 2	P	Novel	c.2843G>A (p.Cys948Tyr)	N.A	9	P	CM992152 (35)
21	FG666	*CRB1*	604210	201253.3	c.2843G>A (p.Cys948Tyr)	NA	9	P	CM992152 (35)	c.2843G>A (p.Cys948Tyr)	N.A	9	P	CM992152 (35)
22	FG789	*CRB1*	604210	201253.3	c.2843G>A (p.Cys948Tyr)	NA	9	P	CM992152 (35)	c.2843G>A (p.Cys948Tyr)	N.A	9	P	CM992152 (35)
23	FG850	*CRB1*	604210	201253.3	c.653-1G>A	m	Intron 2	P	Novel	c.2843G>A (p.Cys948Tyr)	(p)	9	P	CM992152 (35)
24	FG901	*CRB1*	604210	201253.3	c.2843G>A (p.Cys948Tyr)	NA	9	P	CM992152 (35)	c.3110_3143dup (p.Ser1049AspfsTer40)	N.A	9	P	CN205417 (1)
25	FG942	*CRB1*	604210	201253.3	c.2843G>A (p.Cys948Tyr)	NA	9	P	CM992152 (35)	c.2843G>A (p.Cys948Tyr)	N.A	9	P	CM992152 (35)
26	FG979	*CRB1*	604210	201253.3	c.2843G>A (p.Cys948Tyr)	NA	9	P	CM992152 (35)	c.2843G>A (p.Cys948Tyr)	N.A	9	P	CM992152 (35)
27	FG981	*CRB1*	604210	201253.3	c.2843G>A (p.Cys948Tyr)	NA	9	P	CM992152 (35)	c.3827_3828del (p.Glu1276ValfsTer4)	N.A	10	P	Novel
28	FG1004	*CRB1*	604210	201253.3	c.2843G>A (p.Cys948Tyr)	NA	9	P	CM992152 (35)	c.3110_3143dup (p.Ser1049AspfsTer40)	N.A	9	P	CN205417 (1)
29	FG319	*CRX*	602225	000554.6	c.434del (p.Pro145LeufsTer42)	NA	4	LP	CD2033314 (1)	-				
30	FG635	*GUCY2D*	600179	000180.4	c.389del (p.Pro130LeufsTer36)	m	2	P	CD962030 (4)	c.1343C>A (p.Ser448Ter)	p	4	P	CM002036 (5)
31	FG337	*IQCB1*	609237	001023570.4	c.1567+2_*2del	NA	15		Novel	c.1567+2_*2del	N.A	15		Novel
32	FG236	*LCA5*	611408	181714.4	c.1243G>T (p.Glu415Ter)	NA	9	P	CM205420 (1)	c.1569_1582del (p.His523GlnfsTer16)	N.A	9	LP	Novel
	FG237	*LCA5*	611408	181714.4	c.1243G>T (p.Glu415Ter)	NA	9	P	CM205420 (1)	c.1569_1582del (p.His523GlnfsTer16)	N.A	9	LP	Novel
33	FG360	*LCA5*	611408	181714.4	c.1243G>T (p.Glu415Ter)	m	9	P	CM205420 (1)	c.1243G>T (p.Glu415Ter)	p	9	P	CM205420 (1)
34	FG496	*LCA5*	611408	181714.4	c.1243G>T (p.Glu415Ter)	m	9	P	CM205420 (1)	c.1243G>T (p.Glu415Ter)	p	9	P	CM205420 (1)
35	FG600	*LCA5*	611408	181714.4	c.1243G>T (p.Glu415Ter)	NA	9	P	CM205420 (1)	c.1243G>T (p.Glu415Ter)	N.A	9	P	CM205420 (1)
36	FG659	*LCA5*	611408	181714.4	c.1243G>T (p.Glu415Ter)	NA	9	P	CM205420 (1)	c.1243G>T (p.Glu415Ter)	N.A	9	P	CM205420 (1)
37	FG856	*LCA5*	611408	181714.4	c.1243G>T (p.Glu415Ter)	NA	9	P	CM205420 (1)	c.1243G>T (p.Glu415Ter)	N.A	9	P	CM205420 (1)
38	FG851	*LCA5*	611408	181714.4	c.1243G>T (p.Glu415Ter)	NA	9	P	CM205420 (1)	c.1243G>T (p.Glu415Ter)	N.A	9	P	CM205420 (1)
39	FG1002	*LCA5*	611408	181714.4	c.1243G>T (p.Glu415Ter)	NA	9	P	CM205420 (1)	c.1243G>T (p.Glu415Ter)	N.A	9	P	CM205420 (1)
40	FG465	*NMNAT1*	608700	001297778.1	c.769G>A (p.Glu257Lys)	m	5	LP	CM127755 (33)	c.364del (p.Arg122GlyfsTer20)	p	4	P	CD127792 (6)
41	FG787	*NMNAT1*	608700	001297778.1	c.769G>A (p.Glu257Lys)	m	5	LP	CM127755 (33)	c.507G>A (p.Trp169Ter)	p	5	P	CM127758 (9)
42	FG165	*PRPF31*	606419	015629.4	c.1060C>T (p.Arg354Ter)	NA	10	P	CM1310332 (13)					
43	FG454	*RAB28*	612994	001017979.3	c.331_333del (p.Val111del)	m	4	LP	Novel	c.331_333del (p.Val111del)	N.A	4	LP	Novel
44	FG402	*RDH12*	608830	152443.3	c.295C>A (p.Leu99Ile)	NA	5	P	CM042465 (18)	c.295C>A (p.Leu99Ile)	N.A	5	P	CM042465 (18)
45	FG68	*RDH12*	608830	152443.3	c.295C>A (p.Leu99Ile)	m	5	P	CM042465 (18)	c.295C>A (p.Leu99Ile)	(p)	5	P	CM042465 (18)
	FG69	*RDH12*	608830	152443.3	c.295C>A (p.Leu99Ile)	m	5	P	CM042465 (18)	c.295C>A (p.Leu99Ile)	(p)	5	P	CM042465 (18)
46	FG383	*RDH12*	608830	152443.3	c.295C>A (p.Leu99Ile)	NA	5	P	CM042465 (18)	c.295C>A (p.Leu99Ile)	N.A	5	P	CM042465 (18)
47	FG429	*RDH12*	608830	152443.3	c.295C>A (p.Leu99Ile)	NA	5	P	CM042465 (18)	c.295C>A (p.Leu99Ile)	N.A	5	P	CM042465 (18)
48	FG612	*RDH12*	608830	152443.3	c.295C>A (p.Leu99Ile)	NA	5	P	CM042465 (18)	c.295C>A (p.Leu99Ile)	N.A	5	P	CM042465 (18)
49	FG667	*RDH12*	608830	152443.3	c.715dup (p.Arg239ProfsTer34)	NA	8	P	CI118737 (2)	c.715dup (p.Arg239ProfsTer34)	N.A	8	P	CI118737 (2)
50	FG694	*RDH12*	608830	152443.3	c.295C>A (p.Leu99Ile)	m	5	P	CM042465 (18)	c.716G>T (p.Arg239Leu)	p	5	P	CM205421 (3)
51	FG780	*RDH12*	608830	152443.3	c.295C>A (p.Leu99Ile)	NA	5	P	CM042465 (18)	c.295C>A (p.Leu99Ile)	N.A	5	P	CM042465 (18)
52	FG247	*RP1*	603937	006269.2	c.5564del (p.Lys1855ArgfsTer42)	m	4	P	Novel	c.5564del (p.Lys1855ArgfsTer42)	p	4	P	Novel
53	FG514	*RP1*	603937	006269.2	c.5564del (p.Lys1855ArgfsTer42)	m	4	P	Novel	c.5564del (p.Lys1855ArgfsTer42)	p	4	P	Novel
54	FG487	*RPGRIP1*	605446	020366.4	c.1077+1_3100-1dup	NA	10–19		Novel	c.1077+1_3100-1dup	N.A	10–19		Novel
55	FG853	*SPATA7*	609868	018418.5	c.1171C>T (p.Arg391Ter)	NA	11	P	CM1817912 (1)	c.1171C>T (p.Arg391Ter)	N.A	11	P	CM1817912 (1)
56	FG441	*TULP1*	602280	003322.6	c.1149C>A (p.Asp383Glu)	m	12	LP	Novel	c.1149C>A (p.Asp383Glu)	p	12	LP	Novel
57	FG118	*NR2E3*	604485	014249.4	c.932G>A (p.Arg311Gln)	m	6	P	CM000538 (36)	c.932G>A (p.Arg311Gln)	(p)	6	P	CM000538 (36)

**Table 2 ijms-25-06151-t002:** Comprehensive description of clinical findings in the 64 Chilean patients. The table presents individuals’ ophthalmological features, including Best-Corrected Snellen Visual Acuity, cycloplegic refraction, slit lamp biomicroscopy, dilated ophthalmoscopy, digital fundus photography (Optos PLC, Dumferline, Scotland UK), full-field ERG (ERG system, Roland Consult, Wiesbaden, Germany), Spectral Domain Optical Coherence Tomography (SD-OCT) obtained with the Heidelberg Spectralis (Heidelberg Engineering, Heidelberg, Germany) and Fundus Autofluorescence (FAF) (Heidelberg Engineering (Heidelberg, Germany)), depending on the patient age. Abbreviations are as follows: ADRP, Autosomal Dominant Retinitis Pigmentosa; BCVA, Best-Corrected Snellen Visual Acuity; CPA, central and peripheral atrophy; CF, counting fingers; CORD, cone–rod dystrophy; EOSRD, early-onset severe retinal dystrophy; F, female; *F*, coefficient of inbreeding—0 denotes absence of known consanguinity, and 0* indicates highly probable inbreeding due to birth in the island of Chaulinec (Chiloé Archipelago), whose population is limited to 653 inhabitants; ffERG, Full-Field ElectroRetinoGram; GRABSPDVA, generalized RPE atrophy bone spicules, pale disc, and vascular attenuation; GFS, Goldmann–Favre syndrome; HM, hand Motion; LCA, Leber congenital amaurosis; LE, Light Eye; LP, light perception; M, male; m, maternal; (m), maternal inferred; N.A, Not Available; NLP, No Light Perception; NRPD, Nummular RPE Pigmentation Deposit; p, paternal; (p) paternal inferred; PPRPE, Preserved Para-arteriole Retinal Pigment Epithelium; RE, Right Eye; RPE, Retinal Pigment Epithelium; SE, Spherical Equivalent; TRMPC, tapetum reflex mild peripheral changes; YOB, Year Of Birth.

						Disease Symptoms																			
Family	Patient	Gender	YOB	Place of Birth	Consanguinity (*F*)	Age at Presentation	BCVA RE (Decimal)	BCVA LE (Decimal)	Refraction RE (SE)	Refraction LE (SE)	Nystagmus	Oculodigital Sign	Nyctalopia	Photophobia	ffERG Rods	ffERG Cones	GRABSPDVA	CPA	NRPD	PPRPE	TRMPC	Pseudo-coloboma	Others	Initial Diagnosis	Final Diagnosis
1	FG313	M	1996	Chillan	0	9 months	lp	lp	N.A	N.A	Yes	N.A	Yes	Yes	N.A	N.A	Yes	No	No	No	No	No		LCA	LCA
2	FG297	M	1975	Santiago	0.0625	birth	npl	npl	N.A	N.A	Yes	N.A	Yes	No	N.A	N.A	Yes	No	No	No	No	No	Hearing loss, type 2 diabetes mellitus, arterial hypertension, and epileptic seizures	LCA	ALMS
	FG283	M	1973	Santiago	0.0625	birth	lp	lp	N.A	N.A	Yes	N.A	Yes	Yes	N.A	N.A	Yes	No	No	No	No	No	Dense cataract (LE), hearing loss, type 2 diabetes mellitus, arterial hypertension, and epileptic seizures	LCA	ALMS
3	FG277	F	2013	Santiago	0.0625	18 months	0.2	0.2	(+3.25)	(+3.75)	No	No	Yes	No	Rod–Cone	Rod–Cone	No	No	No	No	Yes	No		LCA	LCA
4	FG393	F	2014	Santiago	0	birth	lp	lp	N.A	N.A	Yes	Yes	Yes	No	N.A	N.A	No	No	No	No	Yes	No	Mental impairment; brother affected with syndactyly	LCA	LCA
5	FG50	F	1972	Santiago	0	1 year	lp	lp	(+10.5)	(+10.5)	Yes	N.A	Yes	No	N.A	N.A	No	Yes	Yes	Yes	No	Yes		LCA	LCA
6	FG66	F	2000	Santiago	0	2 years	0.05	0.05	0	0	Yes	No	N.A	N.A	Rod–Cone	Rod–Cone	No	Yes	No	Yes	Yes	Yes		LCA	LCA
	FG224	M	2010	Santiago	0	birth	0.05	0.05	(+12)	(+11.25)	Yes	N.A	Yes	Yes	N.A	N.A	No	Yes	Yes	Yes	No	Yes		LCA	LCA
7	FG112	M	1990	Quipue	0	birth	lp	npl	N.A	N.A	Yes	Yes	Yes	Yes	N.A	N.A	No	Yes	Yes	Yes	No	Yes		LCA	LCA
	FG113	M	2002	Quipue	0	birth	lp	lp	N.A	N.A	Yes	Yes	Yes	Yes	N.A	N.A	No	Yes	Yes	Yes	No	Yes	Keratoconus	LCA	LCA
8	FG128	F	1982	Coinco	0	3 years	cf	cf	N.A	N.A	Yes	Yes	No	Yes	N.A	N.A	No	Yes	Yes	Yes	No	Yes	Keratoconus	LCA, EOSRD	LCA
9	FG239	F	1986	Cañete	0	birth	lp	lp	N.A	N.A	Yes	Yes	Yes	Yes	N.A	N.A	No	Yes	Yes	Yes	No	Yes	Keratoconus	LCA	LCA
10	FG362	M	1969	Angol	0	<1 year	lp	lp	(+5.0)	(+3)	Yes	N.A	N.A	N.A	Abolished	Abolished	No	Yes	Yes	Yes	No	Yes	Vitreous opacities	LCA	LCA
11	FG272	F	1997	Santiago	0	1 year	cf	cf	(+3.75)	(+4.0)	Yes	No	Yes	No	N.A	N.A	Yes	Yes	No	No	No	No		LCA	LCA
	FG365	M	1983	Santiago	0	childhood	lp	lp	impossible	impossible	Yes	Yes	Yes	No	N.A	N.A	N.A	N.A	N.A	N.A	N.A	N.A	White bilateral cataract	LCA	LCA
	FG366	M	1993	Santiago	0	childhood	hm	hm	N.A	N.A	N.A	N.A	Yes	Yes	N.A	N.A	Yes	Yes	No	No	No	No	Keratoconus	LCA	LCA
12	FG390	M	1972	Papudo	0	3 months	npl	lp	N.A	N.A	Yes	Yes	Yes	Yes	N.A	N.A	No	Yes	Yes	No	No	Yes	Keratoconus	LCA	LCA
13	FG432	M	2013	Coihueco	0	6 months	0.2	0.1	(+5.5)	(+6.0)	Yes	N.A	Yes	No	N.A	N.A	No	Yes	Yes	Yes	No	Yes	Type 1 diabetes	LCA	LCA
14	FG436	M	2011	Gorbea	0	3 months	cf	cf	(+8.5)	(+7.5)	Yes	Yes	Yes	Yes	Rod	Cone	No	Yes	No	Yes	No	Yes		LCA	LCA
15	FG444	M	2014	Villarica	N.A	2 months	0.025	0.025	(+8.0)	(+8.0)	Yes	N.A	Yes	Yes	N.A	N.A	No	Yes	Yes	No	No	Yes		LCA	LCA
16	FG456	M	1995	Santiago	0.0313	birth	cf	0.05	(+5.25)	(+4.0)	Yes	N.A	Yes	No	N.A	N.A	No	Yes	Yes	No	No	Yes	Optic nerve drusen	LCA	LCA
17	FG231	F	2010	Chillan	0	7 months	0.15	0.1	(+7.5)	(+7.5)	Yes	No	Yes	No	N.A	N.A	No	Yes	Yes	Yes	No	Yes		LCA	LCA
18	FG395	M	1989	Padre Hurtado	0	birth	hm	cf	NA	NA	Yes	Yes	Yes	Yes	N.A	N.A	No	Yes	Yes	Yes	No	Yes		LCA	LCA
19	FG399	M	2009	Concepcion	0	3 years	0.05	0.15	(−1.75)	(−1.50)	Yes	No	Yes	No	Abolished	Abolished	No	Yes	Yes	Yes	No	Yes	Mild mental impairment	LCA	LCA
20	FG649	M	2003	Quillota	0	birth	0.5	0.6	(−1.50)	(−1.50)	Yes	No	Yes	No	N.A	N.A	No	Yes	Yes	No	Yes	No		LCA, EOSRD	LCA
21	FG666	F	1985	Colina	0	birth	lp	lp	N.A	N.A	No	Yes	Yes	No	N.A	N.A	No	Yes	Yes	No	No	Yes		LCA	LCA
22	FG789	M	2013	Rancagua	0	2 years	0.2	0.2	(+3.00)	(+2.50)	No	No	Yes	No	N.A	N.A	No	Yes	Yes	Yes	No	No		EOSRD	LCA
23	FG850	M	2002	Santiago	0	3 years	0.5	0.2	(−0.50)	NA	No	No	Yes	No	Rod–Cone	Rod–Cone	No	Yes	Yes	Yes	No	No		EOSRD	LCA
24	FG901	F	1988	Antofagasta	0	birth	0.08	0.04	(+0.50)	(+0.75)	Yes	Yes	Yes	Yes	N.A	N.A	No	Yes	Yes	No	No	Yes	Optic disc drusen	LCA	LCA
25	FG942	F	1998	Parral	0	3 years	cf	cf	(+3.75)	(+3.00)	Yes	Yes	Yes	No	Abolished	Abolished	No	Yes	Yes	Yes	No	Yes		LCA	LCA
26	FG979	M	1969	Santiago	N.A	2 years	cf	cf	N.A	N.A	Yes	Yes	Yes	No	N.A	N.A	Yes	Yes	No	No	No	No		LCA, EOSRD	LCA
27	FG981	M	1976	Santiago	N.A	birth	cf	cf	(+3.25)	(+2.50)	Yes	No	Yes	Yes	N.A	N.A	No	Yes	Yes	Yes	No	Yes		LCA	LCA
28	FG1004	F	2018	Santiago	0	birth	0.05	0.05	(+6.5)	(+6.5)	Yes	No	Yes	Yes	N.A	N.A	No	Yes	Yes	Yes	No	Yes		LCA	LCA
29	FG319	F	1991	Santiago	0	4 years	0.025	cf	NA	NA	Yes	N.A	Yes	Yes	N.A	N.A	Yes	Yes	No	No	No	No		LCA	LCA
30	FG635	F	2018	Santiago	N.A	birth	lp	lp	(+4.00)	(+4.00)	Yes	Yes	Yes	Yes	Normal	Cone	No	No	No	No	No	No		EOSRD	LCA
31	FG337	M	2002	Santiago	0	2 months	lp	lp	N.A	N.A	Yes	Yes	No	Yes	N.A	N.A	No	Yes	No	No	No	No	Renal failure at 14 years of age	LCA	SLNS
32	FG236	M	1983	Santiago	0	childhood	hm	hm	N.A	N.A	Yes	N.A	Yes	No	N.A	N.A	Yes	Yes	No	No	No	No		LCA	LCA
	FG237	F	1997	Santiago	0	1 year	cf	cf	N.A	N.A	Yes	No	Yes	No	N.A	N.A	Yes	Yes	No	No	No	No		LCA	LCA
33	FG360	F	1995	Santiago	0.0156	birth	cf	cf	N.A	N.A	Yes	Yes	Yes	No	N.A	N.A	Yes	Yes	No	No	No	No		LCA	LCA
34	FG496	F	1964	Santiago	0	birth	0.05	0.1	(+2.25)	(+2.0)	Yes	Yes	Yes	No	Rod	Cone	No	No	No	No	Yes	No		LCA	LCA
35	FG600	F	1991	Santiago	0	6 months	0.05	0.2	N.A	N.A	Yes	N.A	Yes	Yes	N.A	N.A	Yes	Yes	No	No	No	No		LCA	LCA
36	FG659	M	1999	Santiago	0	birth	0.3	0.4	(+0.50)	(+0.25)	No	No	Yes	No	Rod–Cone	Rod–Cone	Yes	No	No	No	No	No		LCA	LCA
37	FG856	F	1970	Temuco	0.0039	birth	0.1	hm	(−0.75)	(−3.25)	No	No	Yes	Yes	N.A	N.A	Yes	Yes	No	No	No	No		LCA	LCA
38	FG851	M	1989	Santiago	0	birth	0.05	0.05	(−1.00)	(−0.75)	Yes	No	Yes	No	Abolished	Abolished	Yes	Yes	No	No	No	No		LCA	LCA
39	FG1002	F	2016	Iquique	0	3 months	cf	cf	N.A	N.A	Yes	No	Yes	Yes	N.A	N.A	No	No	No	No	Yes	No		LCA	LCA
40	FG465	M	1988	Santiago	0	birth	lp	lp	N.A	N.A	Yes	Yes	No	No	N.A	N.A	N.A	N.A	N.A	N.A	N.A	N.A	White bilateral cataract	LCA	LCA
41	FG787	M	2020	Talca	N.A	birth	N.A	N.A	N.A	N.A	Yes	Yes	Yes	No	N.A	N.A	No	Yes	No	No	No	Yes		LCA	LCA
42	FG165	F	1950	Santiago	0	7 years	lp	lp	N.A	N.A	Yes	N.A	Yes	Yes	N.A	N.A	Yes	Yes	No	No	No	No		LCA	ADRP
43	FG454	F	1964	Los Angeles	0	16 months	0.1	0.1	N.A	N.A	Yes	N.A	No	Yes	N.A	N.A	Yes	Yes	No	No	No	No		LCA	CORD
44	FG402	F	2000	Santiago	0	birth	0.2	0.2	(−0.25)	(+0.25)	No	No	Yes	Yes	Abolished	Abolished	Yes	Yes	No	No	No	Yes	Diffuse paravenous pigmentation	LCA	LCA
45	FG68	M	1986	Requinoa	0	N.A	0.1	0.1	(+0.75)	(+0.75)	No	No	No	Yes	Abolished	Abolished	Yes	Yes	No	No	No	No		EOSRD	LCA
	FG69	M	1986	Requinoa	0	childhood	0.1	0.1	(+1.25)	(+1.5)	N.A	N.A	Yes	No	Abolished	Abolished	Yes	Yes	No	No	No	No		EOSRD	LCA
46	FG383	F	2009	Santiago	0	2 years	0.2	0.1	(+1)	(+1.5)	N.A	N.A	Yes	Yes	N.A	N.A	Yes	Yes	No	Yes	No	No		LCA, EOSRD	LCA
47	FG429	M	1986	Santiago	0	5 years	hm	hm	(−3.0)	(−3.75)	No	No	No	Yes	N.A	N.A	Yes	Yes	No	No	No	No		EOSRD	LCA
48	FG612	F	1988	Machali	0	7 years	npl	hm	N.A	N.A	No	No	Yes	No	N.A	N.A	Yes	Yes	No	No	No	Yes		EOSRD	LCA
49	FG667	F	1996	Santiago	(0.0156)	birth	hm	0.08	N.A	N.A	Yes	No	Yes	No	N.A	N.A	Yes	Yes	No	No	No	Yes		LCA	LCA
50	FG694	F	2011	Arica	0	birth	0.6	0.4	(+1.25)	(+1.25)	Yes	No	Yes	Yes	Abolished	Cone	Yes	Yes	No	No	No	No	Diffuse paravenous pigmentation	LCA	LCA
51	FG780	M	2016	Santiago	0	3 years	0.5	0.5	(+1.25)	(+1.25)	No	No	Yes	Yes	N.A	N.A	No	Yes	No	Yes	Yes	No	Diffuse paravenous atrophy	EOSRD	LCA
52	FG247	F	1988	Santiago	0.0625	childhood	hm	hm	(−9.5)	(−10.5)	Yes	No	Yes	Yes	N.A	N.A	Yes	Yes	No	No	No	No		LCA	LCA
53	FG514	M	1993	Santiago	0*	childhood	0.33	hm	(+0.75)	N.A	Yes	No	Yes	No	N.A	N.A	Yes	Yes	No	No	No	No	Optic disc drusen	LCA	LCA
54	FG487	M	1995	Coquimbo	0	5 years	hm	cf	0	(−3.0)	Yes	No	Yes	No	N.A	N.A	Yes	No	No	No	No	No		LCA	LCA
55	FG853	F	1980	Santiago	0	NA	nlp	lp	N.A	NA	Yes	No	Yes	No	N.A	N.A	Yes	Yes	No	No	No	No		LCA	LCA
56	FG441	F	2018	Santiago	(0.0625)	2 years	cf	hm	(−6.0)	(−5.5)	Yes	No	Yes	Yes	N.A	N.A	Yes	No	No	No	No	No	Persistent ductus arterioso	LCA	LCA
57	FG118	F	2013	Puerto Montt	0	birth	N.A	N.A	N.A	N.A	N.A	N.A	Yes	No	N.A	N.A	No	Yes	Yes	No	Yes	No	Bilateral retinal detachment	LCA	GFS

### 2.3. Subjects Carrying Mutations in Other IRD Genes

Among the 61 resolved individuals with non-syndromic LCA or ESORD, six unrelated subjects harbored variants in genes associated with other retinal diseases (Table 1).

*ADAM9*. We identified homozygosity for an unreported *ADAM9* deletion encompassing exons 5 to 11 (26 to 40 kb in size) in a male evaluated at 9 months of age due to profound visual deficiency with nystagmus, indicative of LCA; however, ERG was not available for diagnosis. At the latest examination (age 26), the patient exhibited nystagmus, photoaversion, night blindness, profoundly impaired vision (LP), and widespread pigmentary retinopathy with central preservation evident at the fundus (Table 2). This phenotype is more severe than the typical *ADAM9*-associated presentation, characterized by poor vision in childhood without nystagmus and photoaversion [28].

*RP1.* Two seemingly unrelated individuals were identified as homozygous carriers of the previously unreported *RP1* 1-bp deletion (c.5564del (p.Lys1855Argfs*42)), affecting the C-terminal region of the protein. We assessed the potential kinship between these individuals by calculating the KING-robust kinship coefficient using the KING Toolset and panel-wide SNP genotypes. The resulting negative coefficient (−0.1297) indicated a non-family relationship. *RP1* truncating mutations have been associated with autosomal dominant or recessive diseases, with the manifestation depending on the location of the variant. Dominant variants affect the middle of the protein and cause RP, while recessive mutations are found in the N- and C-terminal regions and result in more variable and severe phenotypes, including RP, macular dystrophy, cone–rod dystrophy (CORD), and ESORD or LCA [29]. In line with these associations, both subjects received a diagnosis of LCA during early childhood. At their most recent examinations at ages 29 and 34, they exhibited nystagmus, severely impaired vision (HM and a BCVA of 0.33 and bilateral HM, respectively), widespread pigmentary retinopathy with macular atrophy. Additionally, the elderly individual presented with symptoms of photophobia and high myopia (Table 2).

*NR2E3.* Homozygosity for the founder *NR2E3* c.932G>A (p.Arg311Gln) mutation, known to cause Goldmann–Favre syndrome (GFS, MIM#268100), enhanced S-cone syndrome (ESCS, MIM# 268100), and autosomal recessive or dominant RP (RP37, MIM# 611131), was identified in a female patient initially addressed for LCA (Table 2). Following in-depth discussions with the patient prompted by the molecular results, it came to light that she had undergone bilateral retinal detachment and vitrectomy at birth, providing strong indications for a diagnosis of GFS rather than LCA.

*RAB28*. Homozygosity for a likely pathogenic *RAB28* 3-bp deletion predicting the loss of the highly conserved Valine 111 (c.331_333del; p.Val111del) has been identified in an individual with a history of visual deficiency since the age of 16 months. At 55 years, he exhibited bilateral low BCVA of 0.1, along with nystagmus, photoaversion, widespread pigmentary retinopathy, and macular atrophy at the fundus (Table 2). *RAB28* mutations are associated with childhood-onset CORD18 (MIM# 615374), which manifest following reduced BCVA, dyschromatopsia, bull’s eye maculopathy, foveal hyperpigmentation, peripapillary atrophy, extinguished photopic ERG responses, and reduced scotopic ERG responses [30]. Considering the absence of initial ophthalmological data, especially ERG recording and color vision, the diagnosis of CORD over LCA might be taken into account.

*PRPF31*. We identified the recurrent *PRPF31* variant c.1060C>T (p.Arg354*), known to cause autosomal dominant RP [12], in a 72-year-old individual. This person exhibited nystagmus, night blindness, photophobia, and LP, along with widespread pigmentary retinopathy at the fundus (Table 2). Interestingly, the patient had received an LCA diagnosis at the age of seven years. This finding aligns with two unrelated Chinese LCA subjects carrying the same mutation [31,32].

### 2.4. Individuals Who Developed Additional Symptoms Consistent with a Syndromic IRD

*ALMS1*. We identified homozygosity for a novel *ALMS1* 1-bp deletion (c.1092del; p.Asp365Ilefs*11) in two brothers initially addressed for profound visual deficiency with nystagmus near birth. These individuals later developed additional features, including hearing loss, type 2 diabetes mellitus, arterial hypertension, and epileptic seizures, alongside early-onset and severe visual disease, providing further evidence for ALMS. At their respective examinations at ages 47 and 49, they displayed profoundly impaired vision (LP and NLP with photophobia, respectively), along with widespread pigmentary retinopathy at the fundus (Table 2).

*IQCB1*. We discovered homozygosity for a novel deletion in the last exon (exon 15) of the *IQCB1* gene in the sporadic SLNS case included in this study. The individual presented severe visual deficiency from birth with nystagmus, digito-ocular signs of Franceschetti, photophobia, and LP. Renal symptoms emerged at the age of 14, and by 17, the fundus examination revealed macular and peripheral atrophy (Table 2), consistent with the characteristic features of *IQCB1*-associated disease.

## 3. Discussion

Rare hereditary diseases can vary dramatically in prevalence, locus, and allelic heterogeneity depending on geographic region. This study aimed to determine the genetic architecture of severe pediatric IRDs in Chile by studying a cohort of individuals from across the country recruited between 2016 and 2022 at the Hospital El Salvador in the Capital City of Santiago. Sixty-seven individuals from 60 families were included. All were initially seen for a severe visual dysfunction consistent with a provisional clinical diagnosis of LCA or EOSRD.

In two of these families, the diagnosis was secondarily reclassified as syndromic ciliopathy.

Molecular testing of a large panel of candidate genes yielded a 95% molecular diagnosis. Consistent with the typical recessive transmission of severe pediatric retinal dystrophies, biallelic variants were detected in 96.5% of the resolved families (55/57). Homozygosity was observed in two-thirds of these families, supporting high levels of inbreeding in the Chilean population.

In nearly 90% of solved families (49/55, excluding syndromic ciliopathies), the molecular diagnosis was consistent with the clinical diagnosis of LCA/EOSRD. This high diagnosis pick-up rate was associated with a limited locus heterogeneity. Only ten among the twenty-some *LCA* genes were involved. *CRB1* and, within a distance, *RDH12* and *LCA5* were most prevalently implicated (44.4% and 14.3% each, respectively). The very high prevalence of *CRB1* variants in the Chilean pediatric IRD population is mostly due to the recurrence of the p.Cys948Tyr and p.Ser1049Aspfs*40 variants, which together represent 85.7% (48/56) of mutant *CRB1* alleles. The genome of Chilean individuals shows a mixture of European (57.2%), Native American (38.7%), and African (2.5%) ancestry [33]. *CRB1* variants are rather prevalent in Europe and the leading cause of severe pediatric IRDs in Spain, the largest source of European immigration to Chile [34]. Interestingly, the Cys948Tyr substitution that is the most prevalent disease allele in Chile (31/110, 28.2%, of all LCA alleles; 31/56, 55.4% of mutant *CRB1* alleles) is described as the most frequent *CRB1* mutation in Spain (22% of *CRB1* alleles) [13]. This mutation most likely arrived in Chile through one or more European (Spanish) ancestors. The second most prevalent *CRB1* change, p.Ser1049Aspfs*40, representing 30.4%(17/56) of *CRB1* alleles, has not been described previously, with the exception of a unique Chilean family [35]. Based on haplotype similarities among carriers, it is predicted that this mutation is the result of a founder effect that occurred in Chile 11 generations ago (CI: 7–9). Both *CRB1* variants were most likely spread by inbreeding. The remaining six unique *CRB1* disease alleles identified in the Chilean cohort were less frequent (one or two families). The p.Arg764His is the only known variant reported in populations from the Mediterranean region, including Spain, France, Tunisia, Turkey, and Brazil (Table 1). The other substitutions have not been reported previously and, in the light of their scarcity, may have occurred more recently than the p.Ser1049Aspfs*40 change.

The prevalence of *RDH12* variants in the Chilean population is supported by the recurrence of the p.Leu99Ile allele (15/18, 83.3%), which was possibly introduced to Chile through Spain, where it is described as the most frequent *RDH12* mutation [36]. In contrast, although *LCA5* mutations have been described as more frequent in Spain than in other countries [37], the prevalence of this gene in the Chilean population of pediatric IRDs is based on the p.Glu415* nonsense change (16/18, 88.9% of LCA5 alleles), identified solely in Chilean families [35]; this study] and which likely occurred five generations ago (CI: 2–21) in Chile.

Consistent with the high prevalence of *CRB1* p.Cys948Tyr and p.Ser1049Aspfs*40, *RDH12* p.Leu99Ile, and *LCA5* p.Glu415* mutations, the allelic diversity in the Chilean population of pediatric IRDs was even more limited than the locus heterogeneity. Together, these four variants contribute to the disease in more than three-fourths of the families (38/49, 77.6%).

The 20 other LCA-causing variants were private (one or two families; CRX excluded) and again largely homozygous. Ten of them were reported in patients from Europe and hence were likely inherited from European individuals [17,18,21,38]. The remaining nine variants have not been reported yet (Table 1). Three of them were detected in unique families and may have emerged very recently. The three other variants were each identified in two unrelated families in compound heterozygosity, suggesting that they occurred earlier and have begun spreading in Chile.

Interestingly, certain genes commonly associated with Leber congenital amaurosis/early-onset severe retinal dystrophy (LCA/EOSRD) in Western Europe, such as *CEP290* and *RPE65*, were only minimally (*CEP290*) or not at all (*RPE65*) involved in the Chilean cohort. While it would be interesting to determine if this trend extends to the Spanish population, currently, to the best of our knowledge, the exact prevalence of these genes in the Spanish population with LCA is unknown due to limited molecular studies employing mutation-specific screenings [39,40]. Providing a final clinical diagnosis in pediatric retinal diseases can be difficult due to the need for sophisticated ophthalmological explorations and long-term follow-up of extraocular functions. Furthermore, the phenotype and genetic cause of LCA largely overlap those of other IRDs. Therefore, it may not be surprising that we found six subjects (6/64, 9.4%) carrying pathogenic variants in non-typical LCA/EOSRD genes (*ADAM9*, *RP1*, *NR2E3*, *RAB28*, and *PRPF31*). We reassessed the clinical data of these patients and revisited the initial diagnosis in Goldmann–Favre syndrome, cone–rod dystrophy (CORD), and autosomal dominant RP in three of them, consistent with their *NR2E3*, *RAB28*, and *PRPF31* genotypes, respectively.

Contrary to this, the diagnosis of the three individuals with mutations in *ADAM9* (one case) and *RP1* (two cases) was consistent with a disease within the spectrum of EOSRD/LCA. To date, biallelic *ADAM9* mutations have primarily been implicated in childhood-onset cone–rod dystrophy (CORD9), with the singular exception of an LCA subject harboring a homozygous truncating variant [41]. In the case of *RP1*, pathogenic variants have predominantly been associated with dominant RP and, to a lesser extent, recessive RP. More rarely, biallelic *RP1* mutations have been reported in LCA or EOSRD [29,42]. Our study complements earlier reports, reinforcing the involvement of *ADAM9* and *RP1* as uncommon causes of LCA/EOSRD.

Remarkably, the two apparently unrelated individuals associated with *RP1* shared homozygosity for the same novel mutation, raising questions about a potential kinship. However, whole-panel genotype analysis using the KING Toolset did not support such kinship, suggesting the possibility of a highly distant relationship or an independent occurrence at a mutation hotspot, necessitating further analysis.

Finally, consistent with their symptoms, three individuals presented mutations in *ALMS1* (two cases) and *IQCB1* (one case). Early diagnosis of ciliopathies is important but challenging because many phenotypes do not occur in early childhood but develop later on. Early molecular testing can certainly help anticipate the emergence of systemic clinical manifestations that could be underdiagnosed, as families typically seek medical attention when symptoms worsen.

## 4. Families, Materials, and Methods

### 4.1. Subjects and Clinical Assessment

Sixty-seven subjects from 60 Chilean families originating from 26 different cities spanning 12 out of the 16 administrative regions of the country were recruited for LCA or EOSRD at the Department of Ocular Genetics of the Hospital del Salvador in Santiago, Chile, between January 2018 and January 2022. The patients (mean age 29 ± 16 years; range 2–72; median 25.5) underwent a detailed medical history and, when feasible, a full ophthalmologic examination. The visual acuity (VA) was measured using a standardized Snellen chart (in decimals). Patients with very low vision were classified using the semi-quantitative scale “counting fingers” (CF), “hand motion” (HM), “light perception” (LP), and “no light perception” (NLP). The familial medical history was systematically recorded by interviewing the patients and/or parents and drawing a pedigree. The diagnosis was reviewed in three individuals from two families who had further developed extraocular symptoms consistent with Alström (ALMS, MIM# 203800) and Senior Loken (SLNS, MIM# 266900) syndromes, respectively.

Genomic DNA from the patients and family relatives was extracted from peripheral blood using standard protocols or from saliva samples using Oragene-DNA (OG-500) Kit, according to the manufacturer protocol (DNA Genotek, Stittsville, ON, Canada).

This study was approved by the local bioethics committee and by the Comité de Protection des Personnes Ile–de–France II Institutional Review Board (CPP:2015-0303/DC2014-2272). Informed consent adhering to the tenets of the Declaration of Helsinki was received from all participants or their legal guardians.

### 4.2. Capture Panel Design and Library Preparation

A custom panel of 212 IRD genes was used, which includes non-syndromic and syndromic LCA and EOSRD genes and some differential diagnoses of causal genes (Appendix A). Libraries were generated from 2 μg of genomic DNA using the SureSelectXT Library Prep Kit (Agilent, Garches, France). Regions of interest were captured by hybridization using biotinylated complementary 120-bp RNA baits designed with the SureSelect SureDesign software and sequenced on an Illumina HiSeq2500 HT system (Illumina, Evry-Courcouronnes, France) to generate 130-bp paired-end reads with a minimum read depth of 200X.

### 4.3. Bioinformatic Analysis

Sequences were mapped on the hg19 build of the reference human genome (GRCh37) using the Burrows–Wheeler Aligner (BWA) [43] algorithm. Downstream processing was carried out with the Genome Analysis Toolkit (GATK) [44] SAMtools [45]. Single-nucleotide variants and indels were called using the GATK Unified Genotyper based on the 72nd version of the ENSEMBL database. Copy number variations (CNV) were called using a specifically designed algorithm integrated into the Imagine PolyDiag interface [46]. Namely, copy numbers were given by the relative read count for each targeted region, determined by the ratio of the read count for that region divided by the total absolute read counts of all targeted regions of the design. The ratio of the relative read count of a region in a given individual over the average relative read counts in other individuals of the run provides an estimation of the copy number for that region in that individual (method adapted from Goossens et al. [47]). Gene variations were filtered using the PolyDiag interface for rarity and pathogenicity scores according to a large number of public and in-house prediction software available through the PolyDiag Interface [46]. The pathogenicity of variants was interpreted in accordance with the American College of Medical Genetics (ACMG) guidelines by using VarSome 11.15 version [48].

### 4.4. Sanger Validation and Segregation Analysis

The presence of SNV and indels and their segregation with the disease were verified by Sanger sequencing using intronic primers (Appendix A) and the BigDye^®^ Terminator v3.1 on an ABI 3500XL Genetic Analyzer (Applied Biosystems, Thermo Fisher Scientific, Courtaboeuf, France). Data were analyzed using the ABI Sequencing Analysis 6 Software.

### 4.5. Haplotype Analysis

SNPs around the *CRB1* c.3110_3143dup (p.Ser1049Aspfs*40) and *LCA5* c.1243G>T (p.Glu415*) variants were extracted from sequencing datasets and phased with the SHAPEIT2 software [49] to construct haplotypes. Common haplotypes among the subjects carrying the variants and flanking SNPs were then used to estimate the age of the most recent common ancestor using the ESTIAGE software [50], which implements a likelihood-based method. We used allele frequencies and genetic distances (cM) obtained from the 1000 Genomes Phase 3 data [51]. Positions absent from this map were interpolated. For simplicity, we considered a mutation rate of 0 at each marker taken into account in the model.

### 4.6. Assessment of the Potential Shared Ancestry among Individuals with the RP1 c.5564del (p.Lys1855Argfs*42) Variant

We employed the KING Toolset [52] to assess the genetic relatedness among individuals harboring the *RP1* c.5564del (p.Lys1855Argfs*42) variant in homozygosity. The KING Toolset utilizes genome-wide genetic markers to calculate the KING-robust kinship coefficient estimator, which systematically yields negative estimates for unrelated pairs with distinct ancestry. In this analysis, calculations were performed using the genotypes of 7700 SNPs from the panel.

## 5. Conclusions

We demonstrate that *CRB1* is the most frequently mutated gene in pediatric retinal dystrophies in Chile, and we disclose a high degree of inbreeding in affected families, which results in a very limited number of mutations. This may certainly be taken into consideration by health authorities when implementing cost-effective molecular diagnosis of pediatric retinal diseases as well as focusing therapeutic efforts.

## Figures and Tables

**Figure 1 ijms-25-06151-f001:**
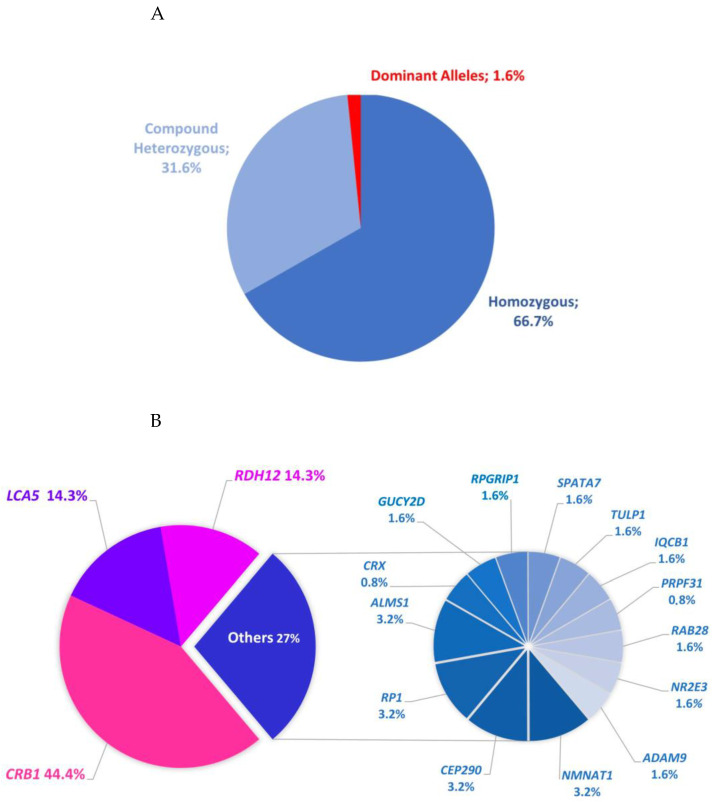
Genetic basis of LCA/EOSRD in the Chilean cohort: (**A**) Proportional identification of homozygous alleles versus compound heterozygous and dominant variants within 64 diagnosed cases. (**B**) Frequency of genes identified in 64 LCA/EOSRD cases. Mutations were identified in 17 genes, with 73% of the mutations found in the top three genes (*CRB1*, *LCA5*, *RDH12*).

## Data Availability

Unreferenced variants have been deposited in the CLINVAR database (submission ID SUB13723950).

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
