# Peer review of "Four Unique Genetic Variants in Three Genes Account for 62.7% of Early-Onset Severe Retinal Dystrophy in Chile: Diagnostic and Therapeutic Consequences"

_ijms, 2024, doi:10.3390/ijms25116151_

Round 1

Reviewer 1 Report

Comments and Suggestions for Authors

Review of “Four unique genetic variants in three genes account for 62.7% of paediatric retinal blindness in Chile: diagnostic and therapeutic consequences” by Moya et al.

The study aimed to determine the genetic cause of severe pediatric IRDs in Chile. The authors demonstrated the impressive diagnostic rate exceeding 95%. They identified the most common genes causative of LCA and EOSRD, namely CRB1, RDH12 and LCA5. Age of causative variants was estimated as well as founder effect for one CRB1 variant was strongly suggested. The authors use next generation sequencing by targeting several retinal genes, the results were analyzed by a variety of computational tools and variants were evaluated in accordance with ACMG guidelines. Furthermore, the authors provide phenotype descriptions for different genetic causes. The manuscript is well structured and well written.

Minor comments:

Line 51 - omit  ”large” cohort. One can argue what is large.

Line 100-101 - What version of Varsome was used? Provide a reference.  

Line 102 – omit a dot in the end of  “2.4. Sanger validation and segregation analysis.”

Line 155- PolyDiag is in -house developed tool, could you possibly provide a reference or an earlier citation

Line 165 – use gnomAD instead of genomAD

Line 191 - consider revision of “before age 1”

Line 395 - finish the sentence, use a bracket and a dot in “(31/110, 28.2%, of 396 all LCA alleles; 31/56, 55.4 % of mutant”

Line 401 – use a dot in “The second most prevalent CRB1 change, p.Ser1049Aspfs*40, representing 30.4%”

Line 402 – revise (”17/56) of CRB1 alleles”

Line 416-417 – revise the sentence and omit own citation

Line 428 – Replace ones with variants in “The three other ones”

Reviewer 2 Report

Comments and Suggestions for Authors

This is a very interesting paper describing the genetic background of early-onset retinal dystrophies in Chilean population.

It presents many interesting facts regarding genetic specificity of population studied, including: 1) predominance of 3 genes and 4 unique pathogenic variants; 2) high proportion of homozygosity suggesting uniform origin of population; 3) high proportion of LCA5 cases that are extremely rare in the other countries; 4) extremely small proportion of dominantly inherited disorders.

Here are my remarks:

- instead of "peadiatric retinal blindness" it would be better to use in the title "early-onset retinal dystrophy";

- LCA is a subgroup of EORD, so in abstract it should be rather LCA/EORD instead of "LCA and EORD";

- it would be great to add some information regarding specific ethnic origin of the families affeted (eg. Andean isolates) and information regarding identified consaquineous matings (with degree of consanguinity)

- I suggest to remove old literature positions (eg. 46,47), especially that they use miccroarray genotyping that in no longer used, and to add some new literature positions that describe the genetic background of LCA/EORD in various populations.
